# Impact of Dosimetric Parameters on Tumor Control in Stereotactic Radiotherapy for Pancreatic Cancer: A Prospective Study on 104 Patients Treated with Simultaneous Integrated Protection (SIP)

**DOI:** 10.3390/cancers17223617

**Published:** 2025-11-10

**Authors:** Marco Lorenzo Bonù, Jacopo Balduzzi, Gloria Pedersoli, Dario Moneghini, Marco Ramera, Nazario Portolani, Jacopo Andreuccetti, Luigi Grazioli, Barbara Frittoli, Sarah Molfino, Anna Maria Bozzola, Maria Teresa Cefaratti, Eneida Mataj, Giulia Volpi, Luigi Spiazzi, Federica Saiani, Alfredo Fiume, Cesare Tomasi, Vittorio Morelli, Paola Vitali, Francesco Frassine, Luca Triggiani, Andrea Guerini, Davide Tomasini, Fabrizia Terraneo, Domenico Della Casa, Fernando Barbera, Stefano Maria Magrini, Michela Buglione

**Affiliations:** 1Radiation Oncology Unit, Istituto del Radio O. Alberti, Spedali Civili Hospital, 25123 Brescia, Italy; marco.bonu@unibs.it (M.L.B.); g.pedersoli005@unibs.it (G.P.); m.cefaratti@unibs.it (M.T.C.); e.mataj@unibs.it (E.M.); cesare.tomasi@live.com (C.T.); v.morelli001@unibs.it (V.M.); paola.vitali@asst-spedalicivili.it (P.V.); francesco.frassine@asst-spedalicivili.it (F.F.); davide.tomasini@unibs.it (D.T.); fabriziaterraneo@virgilio.it (F.T.); barbera.f@tin.it (F.B.); 2Department of Digestive Endoscopy, Spedali Civili Hospital, 25123 Brescia, Italy; dario.moneghini@asst-spedalicivili.it (D.M.); dellacasanico@virgilio.it (D.D.C.); 3Department of Surgery, Spedali Civili Hospital and Brescia University, 25123 Brescia, Italy; marco.ramera@unibs.it (M.R.); nazario.portolani@unibs.it (N.P.); 4Department of Surgery, Spedali Civili Hospital, 25123 Brescia, Italy; jacopo.andreuccetti@asst-spedalicivili.it (J.A.); sarah.molfino@asst-spedalicivili.it (S.M.); 5Department of Radiology, Spedali Civili Hospital, 25123 Brescia, Italy; luigi.grazioli@asst-spedalicivili.it (L.G.); barbara.frittoli@asst-spedalicivili.it (B.F.); 6Department of Pathology, Spedali Civili Hospital, 25123 Brescia, Italy; annamaria.bozzola@asst-spedalicivili.it; 7Radiation Oncology Unit, Azienda Ospedaliera Universitaria Integrata di Verona, 37126 Verona, Italy; giulia.volpi@aovr.veneto.it; 8Department of Medical Physics, Spedali Civili Hospital, 25123 Brescia, Italy; luigi.spiazzi@asst-spedalicivili.it (L.S.); federica.saiani@asst-spedalicivili.it (F.S.); fiumealfredo@yahoo.it (A.F.); 9Radiation Oncology Unit, Spedali Civili Hospital and Brescia University, 25123 Brescia, Italy; luca.triggiani@unibs.it (L.T.); andrea.guerini@unibs.it (A.G.); stefano.magrini@unibs.it (S.M.M.); buglione@med.unibs.it (M.B.)

**Keywords:** pancreatic adenocarcinoma, stereotactic radiotherapy, simultaneous integrated protection

## Abstract

**Simple Summary:**

Treating pancreatic cancer with stereotactic radiotherapy, a precise, high-dose radiotherapy modality, is challenging. The pancreatic gland moves with respiration, and more importantly, the duodenum, stomach, small bowel, and colon are highly sensitive to radiation. As a consequence, treating lesions in contact with such organs is highly complex. Moreover, planning in pancreatic cancer radiotherapy is highly related to physician experience and skill, and limited data are available concerning the major dosimetric variables influencing tumor control. In our prospective, single-arm study, 104 patients were treated with 45 Gy in six fractions, with the simultaneous integrated protection technique to better manage the area of intersection between critical organs and tumor. Our study showed excellent local control with low toxicity. We also identified important dosimetric variables impacting local cancer control, with an area of intersection between tumor and gut receiving a mean dose of at least 28 Gy being related to better local control. Our results may guide the design of future stereotactic radiotherapy trials for pancreatic cancer.

**Abstract:**

**Background**: One of the challenges in treating pancreatic ductal adenocarcinoma (PDAC) with stereotactic radiotherapy (SRT) is to manage lesions abutted to the duodenum, bowel and stomach. Simultaneous integrated protection (SIP) is one of the proposed approaches to increase plan reproducibility and quality. However, no clinical data are available regarding the dosimetric objectives impacting local control probability. **Methods**: This is a prospective, single-arm study. Key inclusion criteria were as follows: PDAC histology; tumor abutment with duodenum, stomach, or small bowel; and SRT schedule consisting of 45 Gy in six fractions. Delineation of the PTV overlapped with critical OARs (PTV_SIP) and PTV outside critical OARs (PTV_Dominant) was mandatory. Dose constraints were as follows: (near) maximum dose, D2cc, and D20cc to critical OARs 38 Gy, 32 Gy, and 24 Gy, respectively. This study was designed to prospectively investigate the main clinical and dosimetric parameters impacting freedom from local recurrence (FFLR). **Results**: From June 2019 to January 2024, 104 patients were enrolled. One-year FFLR was 91.7%. Fifteen events of local failure occurred (17.6%). Mapping of local relapses showed a relapse inside the PTV_SIP area in nine patients and outside the PTV_SIP in six cases (NS). Whole PTV > 69 cc, PTV_SIP > 4 cc, PTV-SIP/whole PTV ratio > 7%, (near) Dmin to PTV_SIP < 25 Gy, mean dose to PTV_SIP < 28 Gy, and (near) Dmin to PTV_Dominant < 29 Gy were associated with worse FFLR. Multivariable analysis showed PTV_SIP absolute volume of more than 4 cc, mean dose to PTV_SIP < 28 Gy and whole PTV > 69 cc were independently related to worse FFLR. One case of acute G4 toxicity and two cases of acute G3 toxicity occurred, with two late toxicity deaths not certainly due to treatment. **Conclusions**: In this prospective study, SIP planning strategy with six fractions is safe and effective in pancreatic targets with critical contact with critical OARs. Given its potential advantages, SIP planning is a potential innovative strategy that should be compared to standard SRT planning in an ad hoc trial design.

## 1. Introduction

Stereotactic radiotherapy (SRT) represents an advanced technique in radiation oncology, enabling the delivery of high biologically effective doses to tumor within a condensed fractionation schedule. In SRT, the therapeutic ratio is optimized by administering highly conformal dose distributions with a steep dose fall-off. The principal objective of SRT lies in the maximization of absorbed doses within the designated target volume while minimizing irradiation of the surrounding normal tissues. In this evolving scenario, the International Commission on Radiation Units and Measurements (ICRU) responded by promulgating ICRU 91 in 2017. ICRU 91 serves as a directive compendium, offering guidance to clinicians and physicists committed to this task [1].

An unresolved challenge in high-dose radiotherapy and specifically in SRT, occurs when organs-at-risk (OARs) are located near or even overlap with the PTV. Some strategies proposed to face this clinical–dosimetric dilemma rely on the reduction in the dose prescribed to the entire planning target volume (PTV) or the intensification of the prescribed dose to the gross tumor volume (GTV), albeit at the expense of reduced PTV coverage [1,2,3].

Another feasible approach to address the aforementioned challenge involves the implementation of the simultaneous integrated protection (SIP) paradigm. The theorization of the SIP concept is rooted in the seminal work by Bunner et al., as detailed elsewhere [4]. In essence, SIP constitutes a theoretical framework wherein the dose limitations of an organ-at-risk (OAR) in proximity to the target dictate the dose prescription within the spatially overlapping region of the target and the OAR. Consequently, SIP serves as a clever methodology to administer the maximum allowable dose to the overlapped region, termed the SIP volume, while concurrently maintaining heightened doses in regions distal to critical anatomical structures [5]. Notably, the SIP concept demonstrates versatility, so it can be applied to both moderately hypofractionated radiotherapy (RT) and various SRT techniques [6,7].

A beneficial aspect of employing the SIP approach lies in the formalization of dosimetric objectives within the overlapping area, allowing quantification and comparison of dosimetric trade-offs between plans [8]. However, a critical consideration in SIP planning involves the uncertainty associated with tumor control in regions subjected to dose reduction. Notably, SIP planning is inherently complex and time-consuming, and it does not provide absolute protection from the potential risk of treatment-related toxicity.

Despite the significant importance of this issue, the huge inconsistency in dose prescription and reporting across published series limits robust conclusions regarding the safety and efficacy of SIP planning [5,9,10,11,12,13,14,15,16,17]. Moreover, ICRU 91 does not specifically address this issue.

Recently, important advances have been made in target volume delineation for pancreatic cancer [18]. Moreover, an in silico multicenter study finally demonstrated the advantages of potential plan consistency and quality achieved with the SIP technique [19]. However, the subsequent understanding of how, where, and how much dose to prescribe in the SIP subvolumes, and how tumor volume parameters and dosimetric parameters impact local control and OAR toxicity, remains a huge area of controversy [20].

For these reasons, we have designed a prospective study encompassing localized PDAC with abutment to the duodenum, stomach, or small bowel, treated with SRT using SIP planning. The objective is to conduct a comprehensive analysis of the clinical–dosimetric determinants influencing freedom from local recurrence (FFLR) when applying the SIP planning technique. Careful follow-up will ensure complete data on acute and late toxicity as well.

## 2. Materials and Methods

This is a prospective, mono-institutional, single-arm study approved by the Ethics Committee (Institutional code NP4904).

### 2.1. Inclusion Criteria

-Age ≥ 18 years old;

-Eastern Cooperative Oncology Group Performance status (ECOG PS) of 0–2;

-For pancreatic adenocarcinoma, multidisciplinary team discussion (MDT) in which the surgical option is excluded for anatomical, biological, or medical reasons (anatomical, biological, or conditional criteria, abbreviated as A, B, and C, respectively);

-Patients with potentially resectable disease may be included after exclusion of surgery in MDT;

-Total blood bilirubin < 2 mg/dL;

-Direct contact with either the duodenum, small bowel, or stomach. Direct contact is defined by a radiologist with 10 years of experience in the upper abdomen and includes direct infiltration of the external layer of an organ. (i.e., lesions with complete infiltration of the duodenal, small bowel, or stomach wall are excluded from the study; baseline endoscopic ultrasound is performed in all patients at the moment of diagnosis and excludes complete infiltration of an organ; if induction chemo with CT scan or patients’ symptoms are suggestive of infiltration, endoscopic ultrasound is performed by protocol before SRT to exclude complete infiltration of the duodenum or stomach). A simplified study flow chart is provided in Appendix A.

### 2.2. Study Procedures

-Patients must be treated with definitive intent on the primary tumor or on a single site of local relapse of disease in the upper abdomen; cN1 disease will be excluded.

-Histological subtypes accepted include primary pancreatic ductal adenocarcinoma or local relapse from pancreatic ductal adenocarcinoma.

-Simulation is performed in the supine position with arms up, laying in the Monarch^®^ (CIVCO^TM^), and completed with three-phase contrast-enhanced CT scan.

-Regarding SRT technique, treatment modalities accepted include Volumetric Modulated Arc Therapy (V-MAT) using Flattening Filter-Free (FFF) beams with 5 mm jaws (Versa HD^®^, Elekta^TM^), Helical Intensity-Modulated Radiation Therapy (IMRT) with FFF beams and 5 mm jaws (Radixact^®^, Accuray^TM^), and robotic techniques utilizing CyberKnife^®^ (Accuray^TM^). Organ motion management includes 4DCT with internal target volume (ITV) creation and abdominal compression, breath-hold (BH) (Active Breathing Coordinator^®^) (Elekta^TM^), and tumor tracking with fiducial markers.

### 2.3. SRT Prescription, Simulation, Planning, and Delivery

#### 2.3.1. SRT Prescription

-The SRT schedule will be 45 Gy in six fractions, delivering at least a biological effective dose (BED) of 78.5 Gy (considering alpha/beta 10 for the tumor). A SIP planning approach is used to manage the dose in the overlap volume between the PTV and dose-limiting OARs such as the duodenum, small bowel, and stomach. A simultaneous integrated boost (SIB) up to 54 Gy in the PTV_Dominant farthest from critical OARs was accepted but not mandatory.

#### 2.3.2. Simulation, Patient Diet, and Intestinal Preparation

-Starting three days before CT simulation and continuing until the end of treatment, a low FODMAP diet and a tablet (or alternatively 50 drops) of Coligas^®^ (Aboca^TM^, Sansepolcro, Italy) per meal, three times a day, are prescribed. In case of serious meteorism detected at CT planning, Elgasin^®^ (Sanitas^TM^, Tortona, Italy) was administered two times a day, at breakfast and dinner, until the end of SRT;

-Patients are simulated and planned by two dedicated radiation oncologists with at least 10 years of experience in the field to ensure consistency in indication, planning, and delivery.

#### 2.3.3. SRT Planning

The procedure for SIP planning followed the seminal paper by Brunner et al.

Specifically, GTV, CTV, ITV, and PTV are defined in accordance with ICRU Reports 50 and 62. Critical OARs for this study are the duodenum, small bowel, and stomach, contoured on each 4D CT phase to obtain their internal margin (IM). The relative planning organ-at-risk volume (PRV) is created by adding 3 mm to each IM. Also, for BH techniques, each critical OAR’s IM is created by using a total of three consecutive scans repeated every 30 s [21]. The PTV is built by geometrically expanding the ITV by 6 mm in the cranio–caudal direction and 4 mm in the radial direction. PTV_SIP is defined as the intersection between the PTV and the PRV of critical OARs, while the rest of the PTV is named as PTV_Dominant. Figure 1 reports an example of planning using the SIP concept.

Elective nodal basin irradiation was not permitted. Inclusion of vessels near or abutted/infiltrated by the tumor was performed by the radiation oncologist even when arterial or venous infiltration was not evident, including at least 180° of the vessel’s circumference in the CTV/ITV.

-Dose constraints for critical organs are reported in Table 1. Specifically, maximum dose (D0.1cc), D2cc, and D20cc to critical OARs were parameters of interest in this present study. A minor violation was defined as a constraint violation not exceeding 2 Gy, while a major violation was defined as any violation more than 2 Gy. Every violation was registered in an ad hoc case report form system (CRFs).

-No oncologic-intent pancreatic surgery is permitted after SRT for patients enrolled.

### 2.4. Follow-Up Protocol and Assessment of Tumor Response

Physical examination, blood count, liver function tests, total bilirubin, INR, ALP, and GGT, are performed 28–30 days after SRT and then at each follow-up visit. Patients are followed every three months during the first year after treatment, every four months for the second year, and every six months thereafter. Assessment of tumor response follows RECIST criteria and is performed with contrast-enhanced CT scan [22]. After the identification of a local relapse based on RECIST criteria, the diagnostic CT images will be exploited to map the site of relapse. The methodology for this analysis will utilize the Velocity^®^ (Varian^TM^) software and will be described thereafter. Tumor markers such as Ca 19.9 and CEA are used to facilitate response evaluation. A contrast-enhanced MRI or a PET scan of the upper abdomen will be performed to confirm tumor response in complex cases.

### 2.5. Study Endpoints

#### 2.5.1. Primary Endpoint Analysis

The pre-specified primary endpoint of the present study is freedom from local recurrence (FFLR). FFLR was defined as the time between the end of SRT and local relapse or the last FU visit. All relevant dosimetric and volumetric plan parameters will be registered as continuous variables and will be exploited to perform a receiver operating characteristic (ROC) curve. The best-fitting value for local relapse will be identified (Yuoden index), and this value will serve as cut-off value in the univariate analysis.

#### 2.5.2. Topographic Analysis of Relapse Site

Moreover, a topographic analysis of sites of local relapse will be performed. Specifically, after a formal diagnosis during per-protocol follow-up by a board-certified radiologist, two dedicated radiation oncologists (i.e., M.L.B. and G.P.), using Velocity^®^ software, will define whether the relapse is inside or outside the SIP area. To do this, rigid coregistration between the planning CT with the radiotherapy plan and follow-up CT/MRI/PET-CT scans will be performed, and the local relapse region of interest (ROI) will be contoured. A relapse will be defined as “occurring in the PTV_SIP area” when at least 0.1 cc of the ROI recurs inside the PTV_SIP volume. On the other hand, a relapse will be defined as “outside the PTV_SIP area” when the entire ROI region is completely outside the PTV_SIP region. Secondary endpoints include acute and late toxicity and their possible relationship with dosimetric aspects of SRT plans, overall survival (OS), and distant metastasis-free survival (DMFS).

### 2.6. Statistical Analysis

The distribution of different clinical and therapeutic features was compared with the Chi-square test. Univariate analysis was performed to identify variables with a statistically significant impact on local control. Survival analysis was performed with the Kaplan–Meier method, and the log-rank test was applied to compare the effects of individual variables on different outcomes with the following variables: age < 73 or ≥73, ECOG PS 0–1 OR 2, tumor stage, pancreatic tumor site (head/uncinate, body, tail), treatment technique, induction chemotherapy yes/no, PTV SIP ≤ 4 cc or >4 cc, minimum dose (Dmin) to PTV_SIP < 25 Gy or ≥25 Gy, mean dose to PTV_SIP < 28 Gy or ≥28 Gy, Dmin to PTV_Dominant < 29 Gy or ≥29 Gy, PTV_SIP/whole PTV ratio < 7% or ≥7%, PTV absolute volume ≤ 68 cc versus more than 68 cc, dose to 95% of whole PTV < 29.8 Gy or ≥29.8 Gy, dose to 95% of PTV_Dominant < 36.8 Gy or ≥36.8 Gy, mean dose to whole PTV < 42 Gy or ≥42 Gy, and mean dose to PTV_Dominant < 45 Gy or ≥45 Gy. Dmin was defined as the dose to 99.9% of the volume (i.e., dosimetric and volumetric plan variables were dichotomized based on ROC curves as previously described).

A Cox proportional hazards model was planned to find independent predictors of local recurrence and survival. Univariate analysis led to the selection of variables that can be considered as predictors (*p* < 0.05 or clinically relevant variables). All tests were two-tailed, and a probability value of less than 0.05 was considered statistically significant. A further analysis using logistic regression will be performed based on multivariate analysis results, with the hypothesis testing a proportional increase in the risk of relapse based on the increase in PTV_SIP absolute volume; in this analysis, the Hosmer–Lemeshow test was applied to the continuous independent variable.

Acute toxicity was defined as any adverse event occurring from the beginning of the RT up to 180 days afterward. Late toxicity was defined as any adverse event occurring at least 181 days after RT (all scored with common terminology criteria for adverse events, CTCAE 5.0). The gathered data were analyzed using SPSS^®^ v.26.0 software (IBM^TM^) and MedCalc^®^v23.4 (MedCalc Software Ltd.^TM^, Ostend, Belgium).

## 3. Results

### 3.1. Description of the Series

Between January 2019 and February 2024, a total of 104 eligible patients were enrolled in this prospective study. The median age of the cohort was 74 years (range 28–93 years).

The predominant ECOG PS scale was 1. Detailed characteristics of the patients are summarized in Table 2.

Twenty-eight patients (27%) were defined by MDT having resectable disease but were excluded from surgery because of medical comorbidity or patient refusal. Sixty-nine patients (66%) did not receive induction chemotherapy due to medical contraindication or poor ECOG PS. Of the 35 patients who underwent induction chemotherapy, the majority were administered Gemcitabine–Nabpaclitaxel for a median of six cycles. A detailed description of MDT decisions regarding contraindication to surgery, CHT, and other clinical variables, such as available baseline Ca 19.9 values is provided in Appendix A. The majority of patients (77%) were treated with LINAC using V-MAT or helical techniques and managed respiratory motion using the 4D CT–Mean CT–ITV technique. Regarding target volumes, the median PTV_SIP volume was 5 cc (range 0.19–135 cc), and the median PTV was 62 cc (range 9.2–190 cc). The median Dmin dose to PTV_SIP was 26 Gy (range 15–40 Gy), and the median Dmin to PTV_Dominant was 30.6 Gy (range 16–44 Gy). The median PTV_SIP/whole PTV ratio was 10% (range 0.4–75%). Protocol dose constraints regarding the maximum dose (D0.1cc), D2cc, and D20cc were met in all cases except three patients, where a minor violation occurred. Table 3 summarizes the characteristics regarding the SRT technique and dosimetry.

### 3.2. Primary Endpoint

After a median follow-up of 19 months (range 6–41 months), 15 events of local failure occurred (15.6%), and 59 patients (61%) were still alive. Mapping of local relapses showed a relapse inside the PTV_SIP area in nine patients and outside the PTV_SIP in six cases. (Chi-square test: non-significant). The rest of the six events of local failure occurred within or at the margins of the PTV_Dominant area.

One- and two-year FFLR was 91.9% and 86.6%, respectively. Median FFLR was not reached. On univariate analysis, absolute PTV_SIP volume more than 4 cc, PTV_SIP/PTV ratio more than 7%, PTV absolute volume more than 68 cc, Dmin to PTV_SIP less than 25 Gy, mean dose to PTV_SIP less than 28 Gy, and Dmin to PTV_Dominant less than 29 Gy were all predictive factors of poorer local control probability (all *p* < 0.05). No other clinical–dosimetric variables impacted local control. Univariate analysis results regarding PTV_SIP absolute volume and mean dose to PTV_SIP and local control are reported in Figure 2a,b. Local control was similar between the three different SRT techniques used in our study.

Multivariate analysis confirmed that PTV_SIP absolute volume of more than 4 cc (odds ratio (OR) 38, 95% confidence interval (C.I.) 4.4–329, *p* < 0.001), mean dose to PTV_SIP < 28 Gy (OR 5.2, 95% CI 1.7–15.8, *p* = 0.003), and whole PTV > 69 cc (OR 4.5, 95% CI 1.1–17.6, *p* = 0.031) were all independently related to an increased risk of local failure. Importantly, χ^2^ did not showed any relevant imbalance between populations concerning other clinical and pathological factors. Results of univariate and multivariate analyses are presented in Table 4 and Table 5.

### 3.3. Secondary Endpoints

Logistic regression was performed defining the independent variable (x) as the absolute volume of PTV_SIP for each patient of the series, and the dual-dependent variable (y) as the event of local control/local failure. Results of logistic regression showed a trend toward decreased local control probability with every single unitary increase in PTV_SIP absolute volume (OR 1.024, CI 0.99–1.05, *p* = 0.061), where 1.024 is the estimated local control odds ratio associated with a 1 cc increase in PTV_SIP absolute volume. Results of the logistic regression analysis are shown in Appendix A.

Median OS and DMFS for the whole series were 36 months and 18 months, respectively. One- and two-year OS were 89% and 60.8%, with one- and two-year DMFS of 62.7% and 36.3%, respectively.

### 3.4. Toxicity

Acute toxicity occurred in 42 patients, with 15 cases of gastrointestinal G2 toxicity, 9 cases of G3 toxicity (8 cases of cholangitis that required hospital admission and 1 case of acute duodenal stenosis), and 2 cases of G4 toxicity (a splenic abscess managed with external drainage in a pancreatic tail cancer infiltrating the splenic hilum, and 1 case of acute kidney artery obstruction not certainly related to SRT). Late toxicity occurred in 21 cases, with 8 cases of G2 and 2 cases of G3 cholangitis recurrences in patients with biliary tree metallic stent, and 2 cases of death not certainly related to treatment (1 case of septic shock due to cholangitis and one due to intestinal ischemia in a patient that experienced acute kidney artery occlusion). In the three cases with dose constraint violations, no acute toxicity greater than G2 or late toxicity more than G1 occurred. Table 6 summarizes acute and late toxicities observed in the whole series.

## 4. Discussion

This prospective study detected clinical–dosimetric determinants of tumor control in SIP planning for pancreatic cancer.

Globally, despite the guidance offered by ICRU 91, there remains a notable lack of consensus regarding the primary dosimetric parameters influencing local tumor control. The existing literature suggests various dosimetric parameters, including (minimum doses, maximum doses, and median or mean doses to PTV, CTV, or GTV, as potential determinants for achieving effective tumor control. Such deliberations remain speculative, awaiting conclusive elucidation in data within the available literature that do not specifically address the management of critically located targets.

The SIP technique, alongside other strategies, such as dose painting to the tumor–OAR interface, has been developed to adjust radiation doses within the overlap regions of the PTV and OAR. In a recent review, one-year FFLR ranged from 67% to 98%, when adopting a planning strategy similar to ours, in 5- or 6-day SRT treatment schedules [5].

However, when the SIP planning strategy is used, the choice of optimal candidates, target coverage objectives, and acceptable doses to PTV, PTV_SIP, and PTV_Dominant are mainly based on expert opinion.

After initial skepticism, in which SIP planning was considered an approach similar to the OAR-prioritization concept, this technique has recently been validated, and its advantages have been proved in multicenter in silico studies. However, the following important clinical question remains open: how dose prescription should be tailored and what dosimetric parameters should be prioritized in the subvolume dose prescription when applying the SIP technique [18,19,20].

Findings from this research hold particular significance in contemporary radiotherapy, shedding light on the ambiguous area of dose prescription for all SIP subvolumes and, more importantly, how such dose prescriptions prioritization could impact local control. The six-fraction schedule, previously tested in the experiences of Comito et al. and Tozzi et al., confirmed safety even with the application of a strict SIP/SIB approach, and despite the slightly lower BED compared to the 50 Gy in the five days schedule, proved highly effective in this context.

Our findings indicate the importance of ensuring a mean dose to PTV_SIP of at least 28 Gy. Other dosimetric parameters that seem to have a relevant impact on local control are as follows: PTV_SIP Dmin above 25 Gy and Dmin to PTV_Dominant higher than 29 Gy.

Our observations highlight the expected link between a larger PTV absolute volume and a larger SIP volume and the reduced likelihood of local control, as follows: the cohort characterized by PTV more than 68 cc and SIP volumes above 4 cc exhibited a significant compromise in tumor control probability. Despite this observation, caution may be adopted to consider this as an absolute threshold warranting the exclusion of patients from the SIP planning strategy. The results of logistic regression analysis suggest a trend toward a gradual reduction in tumor control probability with increasing overlap volume (i.e., PTV_SIP volume), thereby surpassing the simplistic “cut-off” concept elucidated by log-rank and Cox regression results. This observation should be interpreted with caution. The low number of relapse events and the lack of statistical significance clearly suggest to considering it for hypothesis-generating purposes.

Regarding toxicity, the dose constraints adopted were shown to be linked with minimal risk of high-grade acute and late toxicity. This underscores the importance of employing a consistent SRT planning approach, wherein OAR delineation incorporates a robust internal margin contouring strategy for all patients. The two late toxic deaths occurred in frail and vulnerable patients, and no certain correlation to treatment could be shown. Specifically, one patient with pre-existing obstruction in the superior mesenteric artery due to severe vascular disease experienced multiorgan failure 13 months after treatment as a consequence of inferior mesenteric artery obstruction. Re-evaluation of the dose to the artery showed a D0.1cc of 12 Gy and D2cc of 7 Gy; for such reasons, the toxicity was deemed unlikely to be related to SRT. The second G5 toxicity was related to septic shock due to cholangitis that occurred 12 months after SRT.

Given the non-negligible incidence of cholangitis in our series, we hypothesize that a possible contribution to the biliary tract superinfection may be the slight immunodepression related to SRT. This hypothesis led us to test microbiome patterns of change before and after pancreatic SRT in an ongoing trial.

The results of this prospective study reveal a notable rate of tumor control, with 15 instances of local relapse observed overall with a similar topographic distribution inside and outside the PTV_SIP, thereby overcoming one possible pitfall of SIP planning, the hypothetical increase in relapses in the overlap region.

Analyzing local control results from a radiobiological standpoint is intricate, yet several hypotheses can be made. Indeed, a primary concern in SIP planning involves the possibility of increasing relapse rate within the overlap area, potentially nullifying the benefits of tailored dose prescriptions to spare OARs. This hypothesis is sustained by the intuitive notion that “lower doses lead to lower tumor control probabilities.” However, our findings demonstrate comparable relapse rates both inside and outside the PTV_SIP. Furthermore, up to 4 cc of SIP volume, local control outcomes remain similar to those series where more radiosensitive cancers are treated with higher doses in less complex anatomical districts [23,24].

Numerous factors could potentially account for this phenomenon. Firstly, the upper abdomen is a region characterized by significant organ motion. To mitigate this challenge, the majority of patients in our series underwent planning and delivery based on the 4D-ITV concept. Despite meticulous setup and delivery techniques, the inherent nature of 4D planning may not entirely eliminate minor spatial inconsistencies between planning and delivery [25]. This could potentially result in SIP intersecting with the PTV_Dominant volume, leading to increased dose deposition in this area that could not be quantified. However, such inconsistencies could also lead to adverse outcomes, such as geographical miss and heightened acute and late toxicity. Notably, none of these adverse effects were observed in our series. Consequently, we believe that other factors possibly warrant consideration to elucidate this observation.

The bystander effect and immunomodulation within the tumor microenvironment, both modulated by SRT, represent captivating areas of active research. The bystander effect implies the increased death rate among unirradiated cells driven by the irradiation of a specific group of neighboring cells. Experimental evidence supports the existence of this effect even with photon irradiation, related to both direct damage and indirect mechanisms [26].

However, these represent only speculative hypotheses, and the impact of both processes remains to be demonstrated [27].

Our study is subject to several limitations. The modest sample size and the relatively low incidence of local relapses limit our comprehensive analysis of volumetric and dosimetric parameters and their correlation with local control. Despite the study’s prospective design and careful consistency in patient selection, simulation, target delineation, dose planning, and delivery, patients were treated with three different techniques addressing organ motion differently. However, technique had no impact on outcomes, highlighting that consistent planning is more important than the specific technique used in advanced SRT. Concerning a mere evaluation of local control, our series characteristics may in part reflect a relatively well selected group of patients, that may contribute in explaining our fair results. A total of 27% of patients were considered potentially resectable based on anatomic criteria, and no cN+ disease was included. On the other hand, resectable patients enrolled in our study often had a high burden of comorbidity that frequently contraindicated induction chemotherapy. Although univariate analysis seems to not suggest an impact on local control, the enrolment of a group of early-stage patients may partly explain our results compared with other published series focused only on locally advanced disease. Moreover, demonstrating that the SIP planning methodology may improve outcomes is beyond the scope of this study, and the lack of a control group may limit the possibility of suggesting SIP planning as the primary planning methodology when treating localized PDAC.

Notwithstanding these limitations, our findings provide valuable insights into the scenario of high-dose radiotherapy for complex pancreatic targets. We suggest that optimal candidates for SRT with the SIP technique are patients with relatively small volumes of overlap, for whom treatment planning should prioritize maintaining a Dmin to PTV_dominant exceeding 29 Gy, a Dmin to PTV_SIP exceeding 25 Gy, and a mean dose surpassing 28 Gy.

Given the observed relationship between PTV_SIP absolute volume and local control, future research advancements should prioritize the use of advanced techniques and treatment modalities to minimize the overlap between the PTV and dose-limiting OARs. To achieve this goal, exploring alternatives to the 4D-ITV delivery modality, such as the breath-hold technique, advanced online IGRT, and advanced tumor tracking/gating methods including online MR guidance, could be useful. In particular, the SIP planning concept, in our view, will integrate perfectly with MR guidance. MR delivery, along with its capability to reduce ITV and possibly PTV margin, will reduce the SIP absolute volume, thereby reducing the area of dosimetric compromise, while the PTV_SIP and PTV_Dominant area will be used for online planning. Of note, despite all the technical and clinical efforts to minimize the volume of overlap, it should be emphasized that radiation oncologists will always face an inevitable area of compromise between target treating and sparing critical OARs in pancreatic cancer planning, particularly for head and uncinate lesions. For such reasons, intestinal preparation with diet along with anti-meteorism agents and the repurposing radioprotective agents to shield the gut mucosa during irradiation, holds promise for optimizing the therapeutic window [28]. Incorporating all these advancements alongside SIP planning modality could potentially enhance treatment outcomes. Collectively, these incremental improvements could pave the way for the design of modern phase I studies aimed at evaluating dose-escalated regimens using more tailored dose constraints in pancreatic and upper abdominal targets.

## 5. Conclusions

In this prospective study, SIP planning strategy with a six-fraction schedule is safe while maintaining high rates of tumor control in pancreatic targets with critical contact with the duodenum, small bowel, and stomach. A PTV_SIP less than 4 cc, a PTV_SIP Dmean more than 28, and a PTV absolute volume > 69 cc are independently related to better local control. Given its potential advantages, SIP planning is a potential innovative strategy that should be compared with standard SRT planning in an ad hoc trial design.

## Figures and Tables

**Figure 1 cancers-17-03617-f001:**
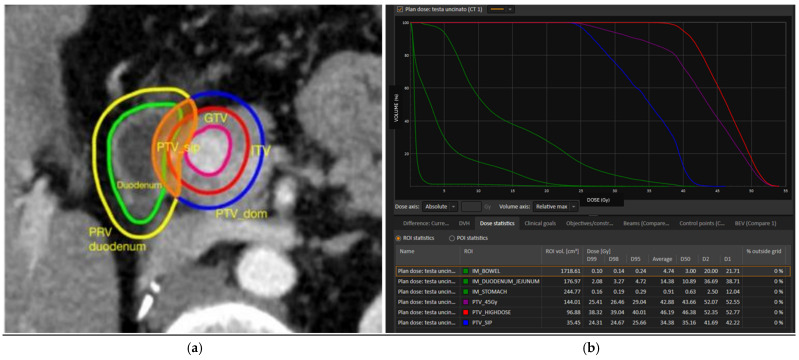
(**a**) Axial CT scan showing an example of the SIP planning technique. PTV_Dominant: planning target volume dominant (blue); PTV_SIP: planning target volume with simultaneous integrated protection (filled orange); ITV: internal target volume (red); and GTV: gross tumor volume (purple). (**b**) Dose–volume histogram showing an example of SIP planning: in green, representation of DVH for the duodenum, bowel, and stomach; in blue, dose to PTV_SIP; in purple, dose to PTV_45 Gy; and in red, dose to PTV_Dominant.

**Figure 2 cancers-17-03617-f002:**
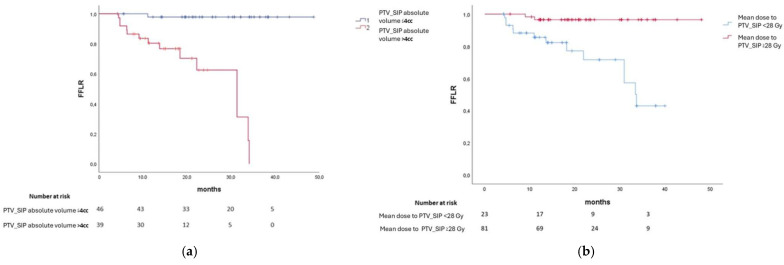
On the left, (**a**) log-rank test showing FFLR for patients having PTV_SIP absolute volume of 4 cc of less versus more than 4 cc (*p* = 0.000); on the right, (**b**) log-rank test showing FFLR for patients with mean dose to PTV_SIP of 28 Gy or less versus more than 28 Gy (*p* = 0.000).

**Table 1 cancers-17-03617-t001:** Dose constraints for dose-limiting organs-at-risk (OARs); dose limits are related to internal margins (IMs) and planning organ-at-risk volume (PRV) of the duodenum, stomach, and small bowel with priority level I for IM and level II for PRV.

OAR	D0.1cc (Gy)	D2cc (Gy)	D20cc (Gy)
IM_Duodenum (Level I priority)	≤38	≤32	≤24
IM_Stomach (Level I priority)	≤38	≤32	≤24
IM_Small bowel (Level I priority)	≤38	≤32	≤24
PRV_IM_Duodenum (Level II priority)	≤40	≤35	≤28
PRV_IM_Stomach (Level II priority)	≤40	≤35	≤28
PRV_IM_small bowel (Level II priority)	≤40	≤35	≤28

Legend: OAR = organ-at-risk; IM = internal margin; and PRV = planning organ-at-risk volume.

**Table 2 cancers-17-03617-t002:** Patients’ characteristics.

Variable	N° (%)
**Age**	Median 74 y (range 28–93 y)
**Gender**	
M	55 (53%)
F	49 (47%)
**ECOG PS**	
0	23 (22%)
1	55 (53%)
2	26 (25%)
**Target lesion**	
Primitive ductal adenocarcinoma	77 (80%)
Local relapse of pancreatic ductal adenocarcinoma	27 (20%)
**Pancreatic subsite**	
Head/uncinate process	50 (48%)
Body	22 (21%)
Tail	5 (5%)
Local relapse	27 (26%)
**Tumor stage (at time of treatment)**	
cT2 cN0	18 (17%)
cT3 cN0	10 (10%)
cT4 cN0	49 (47%)
Local relapse	27 (26%)
**Previous medical treatment**	
No	69 (66%)
Yes	35 (34%)
**Baseline Ca 19.9**	137 U/mL (range 3–1680 U/mL)

Legend: ECOG PS = Eastern Cooperative Oncology Group Performance Status.

**Table 3 cancers-17-03617-t003:** Technical and dosimetric characteristics of the series.

Variable	N° (%)
**Technique**	
V-MAT (FFF)	40 (38.5%)
Helical IMRT	40 (38.5%)
Robotic IMRT	24 (23%)
**Organ motion management**	
4D ITV	82 (79%)
Tumor tracking	15 (14%)
Breath-hold	7 (7%)
**IGRT**	
Fiducial tracking	89 (86%)
CBCT	15 (14%)
**Median PTV_SIP (cc)**	5 cc (range 0.19–135 cc)
**Median PTV (cc)**	62.5 cc (range 9.2–190 cc)
**PTV_SIP/PTV ratio**	Median 10.2%
	(range 0.4–75%)
**(Near) Minimum dose to PTV_SIP**	Median 26 Gy
	(range 14–40 Gy)
**(Near) Minimum dose to PTV_dominant**	Median 30.6 Gy
	(range 15–44 Gy)
**SIB up to 54 Gy in PTV_dominant**	
No	60 (58%)
Yes	44 (42%)
**Dose to 95% of whole PTV**	Median 28.9 Gy
	(range 19–41 Gy)
**Dose to 95% of PTV_Dominant**	Median 36.9
	(range 24.7–50 Gy)
**PTV SIP**	
≤4 cc	45 (47%)
>4 cc	59 (53%)
**Minimum dose to PTV_SIP**	
<25 Gy	45 (47%)
≥25 Gy	59 (53%)
**Mean dose to PTV_SIP**	
<28 Gy	23 (24%)
≥28 Gy	81 (76%)
**Minimum dose to PTV_Dominant**	
<29 Gy	39 (41%)
≥29 Gy	65 (59%)
**PTV_SIP/whole PTV ratio**	
<7%	44 (46%)
≥7%	60 (54%)
**PTV absolute volume**	
≤68 cc	58 (60%)
>68 cc	46 (40%)
**Mean dose to whole PTV**	
<42 Gy	33 (34%)
≥42 Gy	71 (66%)
**Mean dose to PTV_Dominant**	
<45	47 (49%)
≥45 Gy	57 (51%)

Legend: VMAT = volumetric modulated arc therapy; FFF = flattening filter-free; IMRT = intensity-modulated radiotherapy; ITV = internal target volume; CBCT = cone-beam CT; PTV = planning target volume; and SIP = simultaneous integrated protection.

**Table 4 cancers-17-03617-t004:** Univariate analysis results.

Variable	One-Year FFLR	*p*-Value
Age		0.1
<73	86.6%	
≥73	96.5%	
ECOG PS		0.078
0–1	89.3%	
2	98.5%	
Tumor stage		0.298
cT2 cN0	89.1%	
cT3 cN0	100%	
cT4 cN0	100%	
Local relapse	89.4%	
Pancreatic tumor site		0.365
Head/uncinate process	91%	
Body	92.3%	
Tail	85.7%	
Local relapse	91%	
Treatment technique		0.557
V-MAT (FFF)	89.2%	
Helical IMRT	97.4%	
Robotic IMRT	87.3%	
Induction chemotherapy		0.740
Yes	93.4%	
No	88%	
PTV SIP		<0.001
≤4 cc	98%	
>4 cc	85.8%	
Dmin dose to PTV_SIP		<0.001
<25 Gy	85%	
≥25 Gy	96.5%	
Mean dose to PTV_SIP		<0.001
<28 Gy	79.8%	
≥28 Gy	95%	
Dmin dose to PTV_Dominant		0.016
<29 Gy	88.9%	
≥29 Gy	93.6%	
PTV_SIP/whole PTV ratio		0.02
≤7%	95.1%	
>7%	89.6%	
PTV absolute volume		<0.001
≤69 cc	98.2%	
>69 cc	83.3%	
Dose to 95% of whole PTV		0.097
<29.8 Gy	89%	
≥29.8 Gy	91.7%	
Mean dose to whole PTV		0.178
<42	90%	
≥42	93.8%	
Dose to 95% of PTV_Dominant		0.079
<36.8 Gy	87%	
≥36.8 Gy	91%	
Mean dose to PTV_Dominant		0.851
<45	93%	
≥45	90.7%	

**Table 5 cancers-17-03617-t005:** Multivariate analysis results.

Variable	Odds Ratio (95% CI)	*p*
PTV SIP > 4 cc	38	*p* < 0.001
	(4.4–329)	
Mean dose to PTV_SIP < 28 Gy	5.2	*p* = 0.003
	(1.7–15.8)	
Whole PTV > 69 cc	4.5	*p* = 0.031
	(1.1–17.6)	

**Table 6 cancers-17-03617-t006:** Acute and late toxicities of the series.

Acute Toxicity	G1	G2	G3	G4	G0
*Emesis and nausea*	5	3	0	0	0
*Kidney artery occlusion*	0	0	0	1	0
*Biliary tract stenosis*	0	2	0	0	0
*Cholangitis*	0	5	8	0	0
*Gastro-duodenitis*	3	5	0	0	0
*Anorexia*	0	0	0	0	0
*Duodenal stenosis*	0	0	1	0	0
*Diarrhea*	5	0	0	0	0
*Splenic abscess*	0	0	0	1	0
**Total**	17	15	9	2	0
**Late Toxicity**	**G1**	**G2**	**G3**	**G4**	**G5**
*Cholangitis*	0	8	2	0	0
*Gastro-duodenitis*	7	0	0	0	0
*Septic shock*	0	0	0	0	1
*Intestinal* ischemia	0	0	0	0	1
**Total**	7	8	2	0	2

## Data Availability

All anonymized data are available under reasonable and explicit request to corresponding author.

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
