# Peer review of "Impact of Dosimetric Parameters on Tumor Control in Stereotactic Radiotherapy for Pancreatic Cancer: A Prospective Study on 104 Patients Treated with Simultaneous Integrated Protection (SIP)"

_cancers, 2025, doi:10.3390/cancers17223617_

Round 1
Reviewer 1 Report
Comments and Suggestions for Authors
Attached

Reviewer 2 Report
Comments and Suggestions for Authors
This manuscript presents a prospective study on 104 PDAC patients treated with stereotactic radiotherapy using the Simultaneous Integrated Protection (SIP) technique. The authors identify dosimetric thresholds associated with improved local control and demonstrate a safe toxicity profile.
The topic is clinically relevant and the methodology solid, representing one of the few prospective datasets in this challenging anatomical setting. However, the manuscript requires revision in multiple areas before it can be considered for publication.
Major Comments
- Study design and lack of control arm. While the prospective nature is commendable, the absence of a control group (e.g., non-SIP or standard SBRT plans) limits the causal inference regarding the benefit of SIP. This limitation should be clearly emphasized in both the Discussion and Conclusions.
- Technical heterogeneity. Patients were treated with three different delivery systems (VMAT, Helical IMRT, CyberKnife) and varying motion management strategies. Please discuss whether this variability could have influenced dosimetric parameters or clinical outcomes.
- Statistical robustness and dosimetric cut-offs. The dosimetric thresholds (28, 29, and 30 Gy) are derived from an exploratory analysis. These should be explicitly presented as hypothesis-generating rather than definitive. Consider adding a continuous-variable analysis (e.g., logistic regression, ROC analysis) to confirm these trends.
- Toxicity correlation. The manuscript provides detailed toxicity data but does not explore correlations between dosimetric parameters and events of significant grade. Even a brief descriptive or exploratory analysis could reinforce the clinical safety of SIP.
- Overinterpretation of biological mechanisms. The sections on “bystander effect” and “immunomodulation” are speculative and not supported by data from the current study. These should be shortened or clearly labeled as theoretical background only.
Minor Comments
- Ensure consistent terminology between SRT and SBRT throughout the text.
- Clarify who assessed FFLR (treating team vs independent radiologist).
- Figure 1 could better illustrate SIP vs Dominant subvolumes (e.g., include dose wash or DVH examples).
- Consider moving key findings from Supplementary Tables 2–3 into the main manuscript for readability.
- Correct typographical and formatting errors (e.g., “small bowell,” “be er local control,” “permi ed”).
-Tone down assertive language in the Abstract and Conclusions (emphasize that these findings are preliminary and require external validation).
The study is scientifically valuable and addresses an important clinical problem. The SIP concept and the definition of predictive dosimetric thresholds are of potential interest to the SBRT community. However, due to technical heterogeneity, lack of control group, exploratory statistics, and language issues, a major revision is warranted before reconsideration.
Comments on the Quality of English LanguageThe manuscript requires professional English editing. There are frequent minor errors (e.g., be er → better, abu ed → abutted, permi ed → permitted, chemo → chemotherapy).
Syntax and style should be improved for fluency and readability.
Round 2
Reviewer 1 Report
Comments and Suggestions for Authors
Appreciate the revision work!
The manuscript shows clear improvement. However, several areas can be refined further — particularly the title, which feels a bit long and lacks a strong, conclusive theme. Some section headings, subtitles, and paragraph structures could also be streamlined to enhance clarity and flow. Overall, this is a good step forward — well done on the revisions.
best wishes.
Author Response
We thank the reviewer for the valuable feedback.
We have made a slight modification to the title to better emphasize the impact of dosimetric variables without losing any information.
Additionally, we have further divided the Materials and Methods section into subparagraphs dedicated to simulation, patient preparation, planning, SRT delivery, and follow-up, to improve readability.
We hope these minor revisions will enhance the clarity and overall readability of the manuscript.
Reviewer 2 Report
Comments and Suggestions for Authors
The authors have thoroughly addressed all the major and minor issues raised during the previous review round.
The revised manuscript shows substantial improvements in methodological rigor, clarity of presentation, and interpretative balance.
I believe it now meets the standards of Cancers and is suitable for publication pending small editorial proofreading.
Author Response
we thank the reviewer for his feedback
kind regards